# Beyond Transcription: Fine-Tuning of Circadian Timekeeping by Post-Transcriptional Regulation

**DOI:** 10.3390/genes9120616

**Published:** 2018-12-10

**Authors:** Julieta Lisa Mateos, Maria José de Leone, Jeanette Torchio, Marlene Reichel, Dorothee Staiger

**Affiliations:** 1Fundación Instituto Leloir, Av. Patricias Argentinas, Buenos Aires C1405BWE, Argentina; mjdeleone@leloir.org.ar (M.J.d.L.); jtorchio@leloir.org.ar (J.T.); 2RNA Biology and Molecular Physiology, Faculty for Biology, Bielefeld University, Universitaetsstrasse 25, D-33615 Bielefeld, Germany; marlene.reichel@uni-bielefeld.de

**Keywords:** circadian, post-transcriptional, RNA, splicing, RNA-binding proteins

## Abstract

The circadian clock is an important endogenous timekeeper, helping plants to prepare for the periodic changes of light and darkness in their environment. The clockwork of this molecular timer is made up of clock proteins that regulate transcription of their own genes with a 24 h rhythm. Furthermore, the rhythmically expressed clock proteins regulate time-of-day dependent transcription of downstream genes, causing messenger RNA (mRNA) oscillations of a large part of the transcriptome. On top of the transcriptional regulation by the clock, circadian rhythms in mRNAs rely in large parts on post-transcriptional regulation, including alternative pre-mRNA splicing, mRNA degradation, and translational control. Here, we present recent insights into the contribution of post-transcriptional regulation to core clock function and to regulation of circadian gene expression in *Arabidopsis thaliana*.

## 1. Introduction

Higher plants, like most organisms, are exposed to periodic cycles of day and night, caused by the rotation of the earth around its axis. Day and night means the availability or absence of light and ensuing episodes of energy production and energy consumption, recurrent phases of elongation growth, or activity of pollinators. To optimally prepare for the recurrent events, plants rely on an endogenous timekeeper, the circadian clock [1,2]. The core clockwork operates in each cell and consists of a set of clock proteins [3,4]. Through activation and repression of transcription of their own genes, the clock proteins generate a self-sustained 24 h oscillation in clock protein abundance (Figure 1a) [5].

At the molecular level, the core clockwork comprises a set of interwoven feedback circuits. The central loop consists of the two Myb transcription factors LATE ELONGATED HYPOCOTYL (LHY) and CIRCADIAN CLOCK ASSOCIATED 1 (CCA1), peaking at dawn, and the pseudoresponse regulator TIMING OF CAB EXPRESSION 1 (TOC1), peaking at dusk, which reciprocally repress their own expression [6,7,8]. The central loop is interwoven, with two loops active in the morning or the evening. Through the morning loop, LHY and CCA1 activate the expression of the PSEUDORESPONSE REGULATORS (*PRRs*) *PRR9* and *PRR7*, which in turn repress *CCA1* and *LHY*. Recently, by monitoring rapid changes that followed the induction of an inducible *LHY* transgene, a reduction in *PRR7* and *PPR9* levels was observed within 2 h of induction, demonstrating that LHY protein acts as a repressor rather than as an activator of *PRR* genes [9]. A longer time of induction of *LHY* recapitulated the original observations where LHY was placed as an activator of *PPR7* and *PPR9* [10], suggesting that the activator role was possibly due to indirect effects of strong overexpression of the transcription factor. *PRR7* and *PRR9* expression is then switched off during the night through the evening complex (EC), comprising the LUX ARRHYTHMO (LUX) transcription factor and the EARLY FLOWERING 3 (ELF3) and ELF4 proteins [11,12,13]. As a result, *CCA1* and *LHY* transcription is reactivated. In the evening loop, the EC and TOC1 reciprocally regulate their expression. The interconnection of these feedback loops is thought to contribute to the robustness of rhythmic clock gene expression. Apart from regulation at the transcriptional level, post-translational modification of clock proteins and protein–protein interactions are crucial for the clockwork to maintain a 24 h period [14,15].

Rhythmically expressed clock proteins in turn regulate expression of a large fraction of the transcriptome, causing transcripts to cycle with a 24 h rhythm so that they are expressed at the optimal time of day. Enhancer trapping using a promoter-less luciferase reporter in *Arabidopsis* has shown that one-third of the genome is under transcriptional control by the clock [16]. However, activation of the *LIGHT HARVESTING CHLOROPHYLL BINDING PROTEIN* (*LHCB1*3*) promoter was rhythmic, but transcript levels were constitutive, suggesting that changes in mRNA stability in the course of the day obscured rhythmic transcription [17]. Along the same lines, *CATALASE3* mRNA was kept constantly at high levels in constant darkness, whereas *CATALASE3* promoter-driven luciferase activity still oscillated with an evening peak [18,19]. In contrast, *NITRATE REDUCTASE* mRNA oscillated despite a time-of-day independent transcription rate [20]. These findings pointed to a post-transcriptional component in rhythm generation and inferred an important role of RNA-binding proteins (RBPs) in circadian timekeeping [21,22,23,24,25].

## 2. Post-Transcriptional Control of the Circadian Clock by Alternative Splicing and Its Connection with Environmental Responses

Messenger RNAs (mRNAs) traverse a long road in the cell until they reach their final and functional state. After transcription has started, several post-transcriptional processes take place that convert pre-mRNA into mature mRNA (Figure 1b). One of these key mechanisms is splicing, the removal of introns accompanied by the assembly of the remaining exons in consecutive order as they appear. Splicing is accomplished by the spliceosome, a high molecular weight complex consisting of five small nuclear ribonucleoprotein particles (snRNPs) and over 200 additional proteins [26,27]. The snRNPs contain small nuclear uridine-rich RNAs (U1, U2, U4, U5, and U6) and are formed by Sm proteins (U1 to U5) or related Sm-like proteins (Like-SM, LSM) (U6). Still, for most of the genes, not all the introns are spliced or all the exons included, giving rise to different transcript variants coming from the same gene. This alternative splicing (AS) can lead to a premature termination codon (PTC) resulting in the degradation of the PTC-containing isoform via nonsense-mediated decay (NMD). Thus, AS can directly alter transcript levels.

Alternative splicing has been linked to circadian clock control in plants, in part due to the observation that mutants defective in a variety of spliceosome components show defects in circadian behaviors [22,24,28,29]. For example, mutants in *LIKE SM* (*LSM*)4 and *LSM5* have a clock-defective phenotype [30] (Table 1). The lengthening of the circadian period observed in these mutants possibly relies on altered expression and AS of key clock genes such as *CCA1*, *PRR9*, and *TOC1* in *lsm5* and *TOC1* in *lsm4* [30]. Interestingly, the *LSM5* transcript is clock-regulated, implying a role in RNA processing and the adjustment of organisms to periodic changes in the environment. Moreover, PROTEIN ARGININE METHYLTRANSFERASE 5 (PRMT5) represents a very well-documented link between the clock and AS. Arabidopsis *prmt5* mutants show a long period [31,32] due to defective *PRR9* splicing [32]. Like *LSM4* and *LSM5*, *PRMT5* is also circadianly regulated [31,32], and mutants show genome-wide alterations in splicing [32,33]. The molecular activity of PRMT5 involves symmetric dimethylation of arginine residues in its target proteins [34,35]. Previously, it has been shown that spliceosomal Sm proteins are modified by arginine methylation, which acts as a signal for loading the Sms onto small nuclear uridine-rich RNAs (snRNAs) to assemble the small nuclear ribonucleoprotein particles (snRNPs) in mammals [36,37]. In *Arabidopsis*, many PRMT5 methylation substrates are directly involved in splicing, including SmD, SmB, LSM4, and glycine-rich protein 7 and 8 (*At*GRP7 and *At*GRP8) [34]. It is therefore temping to think that arginine methylation of splicing factors through PRMT5 is essential and connects circadian rhythms and splicing.

Besides constituting a contact point between AS and the circadian clock, these and other splicing factors were originally isolated as actors engaged in plant development and environmental and stress responses, such as extreme temperature or salt stress [34,35,54,55,56,57]. This strongly suggests an association between AS, clock, and environmental adaptation. The adjustment of the transcriptome is particularly prevalent in response to environmental changes and stress [58,59]. Whereas variations in ambient temperature modulate many physiological processes in plants, the period of the circadian clock is mostly buffered against variation in ambient temperature, a phenomenon designated as “temperature compensation”. However, abrupt temperature shifts result in the resetting of the circadian clock by changing the phase of circadian oscillations, as do abrupt shifts in light, although it is uncertain the mechanisms by which they are effected. (Figure 1a) [60,61,62]. Moving plants to cold temperatures provokes AS of various clock genes (see later). *CCA1* constitutes one interesting example where functionality of environmentally induced AS of a clock gene has been proven. The alternative-spliced isoform of *CCA1* retains intron 4, causing a PTC (see later). It has been predicted that if translation from this isoform starts from an alternative AUG in exon 5, it recreates the reading frame of the C-terminal to half of CCA1 producing a new protein named CCA1-ß. Though this alternative translation start is not yet proven to occur *in vivo*, Seo et al. overexpressed the truncated CCA1-ß that loses the Myb DNA binding domain but keeps the protein domains required for dimerization, and demonstrated that it, by forming nonfunctional heterodimers, inhibits CCA1 activity. *CCA1* alternative splicing is suppressed by cold augmenting the levels of the active isoform and allowing CCA1 to promote freezing tolerance [63]. It remains to be elucidated *in planta* if the molecular mechanism behind this phenotype relies on the action of the two putative protein isoforms of CCA1 or if it is simply a question of more functional *CCA1* available and in consequence more levels of CCA1. In either case, the differential ratio of *CCA1* isoform under cold is responsible. Still, temperature effects on clock gene transcripts and other mRNAs do not affect AS of all introns, but are specific to certain introns possibly depending on their structure or sequence characteristics, or availability or specificity of splicing factors upon temperature. More generally, many splicing related factors respond to different environmental cues and affect splicing of a variety of clock genes. GEMIN2, a spliceosomal small nuclear ribonucleoprotein assembly factor, modulates low temperature effects on splicing of a subset of genes. *Gemin*2 mutants have had a shorter period of the clock, and the effect of temperature on the circadian period has been strongly enhanced (Table 1) [40]. In addition, splicing of several clock genes has been impaired, resembling the effect of cold treatments in AS [40]. Therefore, GEMIN2 attenuates the effect of temperature on the circadian period in *Arabidopsis*, demonstrating once again that post-transcriptional regulatory mechanisms allow plant adaptation to the environment. Two other splicing factors, SKI-interacting protein (SKIP) and SPLICEOSOMAL TIMEKEEPER LOCUS 1 (STIPL1), are involved in the regulation of the circadian clock as well [39,38]. Both mutants have a long period phenotype of the circadian clock, and temperature compensation is altered in *skip-1*. Aberrant AS of *PPR7* and *PRR9* in *skip-1* and of *CCA1*, *LHY*, *PRR9*, *GI*, and *TOC1* in *stipl-1* contribute to the lengthened period of the clock in each mutant [39,38].

Therefore, AS has been raised as one of the mechanisms involved in the setup of the plant circadian clock, likely contributing to plant adaptation to environmental changes as well.

## 3. Nonsense-Mediated Decay Fine-Tuning Circadian Rhythms

### 3.1. Nonsense Mediated Decay in Circadian Control of Transcript Expression

Alternative splicing of mRNAs can lead to events of unproductive alternative splicing via the introduction of a PTC that gives rise to the production of truncated proteins with altered functions. Given that these aberrant products can have detrimental effects as dominant-negative or deleterious gain-of-function activities, eukaryotic cells have evolved several mRNA integrity surveillance mechanisms [64,65,66]. NMD is a quality-control mechanism that selectively degrades mRNAs containing PTCs [67] and other transcripts with abnormal sequences such as long 3′-UTRs or 3′-UTRs harboring introns [68,69,70,71]. The NMD machinery has been extensively studied in yeast and mammals, and many of the pathway key components have also been identified in plants: UP FRAMESHIFT (UPF)1, UPF2, UPF3, and SMG-7 (but not SMG-1, SMG-5, or SMG-6) [67,72,73,74].

As circadian expression patterns cannot always be explained by transcriptional regulation, degradation of transcripts has been suggested as a possible mechanism in shaping rhythmic expression. This raises the question whether NMD, through the control of transcript stability, could have a role in the post-transcriptional regulation of global circadian rhythms. In *Arabidopsis*, direct evidence that circadian expression of mRNA occurs in part through modulating their stability was found for the *CCR-LIKE* (*CCL*) and *SENESCENCE ASSOCIATED GENE 1* (*SEN1*) transcripts, which are targets of the downstream instability determinant pathway (DST), another mRNA degradation mechanism [75]. A direct impact of NMD on clock physiology has been observed in the filamentous fungus *Neurospora crassa*, where the *prd-6* mutant bearing a mutation in the homologous locus of *UPF1* exhibits a short circadian period and an abnormality in temperature compensation [76,77,78]. The control exerted by UPF1 on the fungus circadian clock involves modulating the splicing of the core clock gene frequency (*frq*) through spliceosome and spliceosome-related serine/arginine rich splicing factors [79]. Taken together, this evidence suggests that performing similar experiments in *Arabidopsis* NMD-defective mutants, such as those carrying the strong allele *upf1–3*, would yield important insights into this regulatory aspect. Unfortunately, *upf1–3* results in lethality caused by a constitutive upregulation of pathogen defense [80,81]. Still, such a phenotype could be rescued by crossing the *upf1* mutant with the defense-defective mutant *pad4*, as shown in Reference [81]. Therefore, *pd4-upf1* double mutants could provide an interesting opportunity for exploring the role of the NMD pathway in the fine-tuning of plant circadian rhythms.

### 3.2. Nonsense-Mediated Decay Modulating the Levels of Core Clock Genes

Besides the evident quality-control function of this pathway, NMD coupled to AS in mammals was also found to have a pervasive role in the regulation of mRNA levels. In particular, AS-NMD of splicing factors led to a decline of the productive protein variant and consequently had an effect on post-transcriptional networks controlled by these splicing factors [82,83]. In *Arabidopsis*, the identification of a cohort of naturally occurring NMD-sensitive AS transcripts spanning a wide range of regulatory processes and pathways has also suggested that AS-NMD is a widespread regulatory mechanism in plants [84]. One of the best examples of this is the feedback loop of the two RNA binding proteins, *At*GRP7 and *At*GRP8.

*At*GRP7 (also known as COLD- AND CIRCADIAN-REGULATED (*CCR*)*2*) and *At*GRP8 (*CCR1*) are a pair of closely related RNA-binding proteins that undergo high-amplitude circadian oscillations with a peak at the end of the light phase [85]. It has been found that *At*GRP7 engages in regulating its own oscillations: Binding of *At*GRP7 to its own pre-mRNA causes AS, leading to a transcript with partially retained intron including a PTC that is degraded via NMD, linking NMD to the circadian system for the first time [41,86]. Genome-wide analysis of *At*GRP7 in vivo targets using individual nucleotide resolution crosslinking and immunoprecipitation (iCLIP) unveiled that *At*GRP7 binds to a suite of circadian transcripts and regulates the amplitude of their transcript oscillations [87,42]. This is in line with the view that *At*GRP7 acts as a molecular slave oscillator, receiving input from the core clock at the transcriptional level, conserving rhythmicity through negative autoregulation at the post-transcriptional level and in turn transducing rhythmicity to downstream targets [88]. 

Furthermore, unproductive isoforms of the two core clock genes *TOC1* and *ELF3* undergo NMD after AS [43,44]. This was demonstrated by determining the relative levels of splice variants of these genes after treatment with the NMD suppressor cycloheximide, as well as in NMD-defective mutants [44]. Previous studies have shown that retention of intron 4 in *TOC1* results in the introduction of a PTC [38,43]. Kwon et al. found that this AS event was partially suppressed when plants were exposed to short-day compared to long-day photoperiods, but was induced in both low and high temperatures. On the other hand, AS of *ELF3* transcripts has produced three alternative isoforms by either the inclusion of an alternative exon within intron 2 or the retention of intron 2 or 3, resulting in the introduction of PTCs at different positions in each new isoform (Figure 2) [38,44]. For this gene, the authors showed that the AS was also suppressed under short-day conditions and was enhanced at high temperatures, but was suppressed at low temperatures and high salinity conditions (Figure 2) [44]. AS leading to NMD also mediates responses of the circadian clock to temperature changes [43]. AS of *CCA1* has given rise to an unproductive isoform retaining intron 4 (CCA1-IR4) that contains a PTC while generating an alternative start site giving rise to an alternative isoform that lacks the DNA-binding domain (Figure 2A). Cold treatments decrease the levels of CCA1-IR4, producing a strong increase in the levels of the functional isoform [58]. Interestingly, this retention event is conserved across species [58], though it needs to be determined if this IR4 response to cold is conserved as well. *LHY* moves in the opposite direction of *CCA1*, with its nonfunctional isoforms augmenting at low temperatures [43]. For *LHY*, three alternative isoforms have been detected, two of them substrates of NMD (Figure 2). Whereas CCA1 and LHY are thought to be mostly redundant in controlling circadian rhythm [89,90], they have different roles in temperature compensation of the circadian clock [91]. This correlates with their behavior regarding AS under low temperatures [43]. Other clock genes such as *TOC1*, *PRR3*, *PPR5*, *PRR7*, *PRR9*, and *ELF3* also change the ratio of their isoforms upon cooling (Figure 2) [43,44].

Taken together, these results represent clear examples of NMD controlling clock gene expression, and once again highlight the prominent role of AS, even the unproductive kind, as one of the major underlying mechanisms connecting environmental signals and circadian regulation. 

## 4. Nuclear Transport as a Mechanism Controlling Clock Genes

The contents of the nucleus are separated from the rest of the cell by the double-lipid bilayer of the nuclear envelope. This requires the existence of a highly elaborate transport system that allows selective exchange of molecules between the nucleus and the cytoplasm. Part of this regulation is through the nuclear pore complexes (NPCs), which are large multiprotein channels that penetrate the nuclear envelope, and through which the majority of nucleocytoplasmic exchange occurs [92]. In plants, the composition of the NPC resembles the metazoan NPC, comprised of nucleoporin proteins [93]. It has been found that these proteins are involved in diverse signaling pathways, including interaction of the plants with microorganisms [94,95,96,97,98,99], hormone signaling [100,101], and abiotic stresses [45,102,103]. However, little is known about the influence of the NPC or nuclear transport on the circadian clock. The paradigmatic case for this is *HIGH EXPRESSION OF OSMOTICALLY RESPONSIVE GENES* 1 (*HOS1*), which was first described as a gene involved in cold responses [103,104] and flowering time control [102]. The HOS1 protein is located in the nuclear envelope and interacts with components of the nuclear pore in *Arabidopsis* [93,102]. HOS1 has been implicated in mRNA nuclear export [45], and therefore it has been speculated that it may play a role in regulating the abundance of transcripts [105]. RNA sequencing analysis has shown that *hos1* mutants exhibit mainly upregulated genes, including those that make up the core circadian machinery such as *PRR5*, *TOC1*, *LUX*, *ELF3*, *ELF4*, and circadianly expressed genes [45]. At the same time, a reduction in rhythmicity and variations in the amplitude of the expression of circadian clock genes has been detected under constant light conditions and has led to a lengthening of the period in several *hos1* alleles [45]. Interestingly, the authors pointed out that the altered expression of these genes was also present in other plant lines in which nuclear pore function was compromised, such as in mutants of *SUPRESSORS OF AUXIN RESISTANCE1* and *LOW EXPRESSION OSMOTICALLY RESPONSIVE GENES 4*. Resembling *hos1*, they exhibit a long-period phenotype that has not been found in mutants of *HASTY* (*HST*), which was proposed as facilitating nucleocytoplasmic transport of microRNAs (miRNAs). Nevertheless, recent data argues against a general involvement of HST in miRNA nucleocytosolic translocation [106]. Altogether, this suggests that mRNA export defects result in period lengthening, pointing to an interplay between nuclear transport and the regulation of the circadian clock.

Moreover, the glycine stretch present in *At*GRP7 shows similarity to M9 domains implicated in nuclear trafficking [107], and it has been shown that *At*GRP7 interacts with the nuclear import receptor transportin both in plants and HeLa cells [108]. Interestingly, *At*GRP7 shuttles between the nucleus and cytoplasm [46,47]. To what extent this shuttling is needed for maintaining rhythms in plants remains unknown.

## 5. Circadian Regulation of Translation

One of the key steps, though often overlooked, in rendering mRNAs ready for translation involves polyadenylation of transcripts. Once transcription terminates, polyadenylation takes place, which comprises the cleavage at the end of the 3′UTR of the pre-mRNAs and the addition of the poly(A) tail to the ends by poly(A) polymerases. In some transcripts, the poly(A) tail can be added to different sites, creating multiple transcripts from the same gene (alternative polyadenylation), similar to what AS does by generating multiple isoforms [109,110]. These steps highly influence mRNA stability, nuclear transport, and translation. At a later step, once in the cytoplasm, shortening of the poly(A) tail by deadenylation begins, mostly as a signal for degradation. Consequently, a reduction of the translation rate is observed. The length of the poly(A) tail is not fixed and therefore constitutes another regulatory layer to direct the fate of the transcripts.

Several decades ago, poly(A) tail length was connected to circadian regulation in mammals [111], with the observation of daily variation in the length of the poly(A) tail of vasopressin messenger RNA [111]. Although this and several other studies suggest an effect of polyadenylation and deadenylation on the rhythmicity of animal transcripts [112], little has been done in plants to address this aspect. In *Arabidopsis*, a subset of poly(A) polymerases [113,114] and deadenylases belonging to the PARN and CCR4/CAF1 deadenylase complexes [115,116] have been found to show a circadian pattern of expression [24]. Whereas this correlates to what has been reported in mice [112], whether rhythmic plant poly(A) polymerases and deadenylases also cause circadian variation in poly(A) tail length of transcripts and if this ultimately contributes to rhythmic expression needs to be studied. This post-transcriptional modification has not been widely studied for clock genes throughout the circadian cycle, but is potentially an important mechanism that may contribute to a typical circadian expression pattern.

Translation itself has also been linked to circadian regulation. In the last years, some genome-wide studies on the translatome and the development of efficient techniques in polysome isolation in *Arabidopsis* have started to shed light on the translation field in plants [49,117,118]. Light has been shown to modulate the translation status of several transcripts in *Arabidopsis* [49]. So far, it is known that translation of the key component LHY is induced by light. This, together with *LHY* transcript repression, could contribute to the narrow peak of the LHY protein at dawn [48].

Circadian expression does not always hold up for protein oscillations. One of the early examples is the F-box protein ZEITLUPE (ZTL), which oscillates with a high amplitude while its mRNA is constitutively expressed [119]. In *Arabidopsis* seedlings, a proteomic study using two-dimensional difference gel electrophoresis (2D-DIGE) revealed that ~30%–40% of rhythmic proteins are derived from nonrhythmic transcripts, indicating that translational and even post-translational control contributes to rhythmic protein abundance [120]. Most of the proteins that oscillate are clustered in pathways such as photosynthesis, hormone signaling, translation, transcription, chaperones, and metabolism, indicating the influence of the circadian clock on the regulation of plant physiology and metabolism. Such a lack of correlation between rhythmic proteins and transcripts has also been observed in animals [121,122]. Two recent papers have shed further light onto the circadian regulation of translation. By analyzing polysome loading in wild-type and overexpressors of CCA1 plants during a day, Missra et al. demonstrated that RNA loading into polysome is cyclic and is controlled in part by the circadian clock [123]. Moreover, clock-associated genes have a cyclic pattern of ribosome loading, but amplitude of the cycle is generally very small compared to what happens to their transcripts. More generally, the study showed that there is no robust phase relationship between peak translation states and peak transcript levels [123]. In the second work, the authors integrated genome-wide transcript studies and protein mass spectrometry to analyze performance of a series of clock mutants (*lhy cca1*, *prr7 prr9*, *gi*, and *toc1*) either at the end of the night or at the end of the day. Interestingly, each mutant affected a specific subset of RNAs and proteins, suggesting that the clock acts in a modular mode. Again, most of the transcripts patterns did not correlate with protein levels. Parallel analysis of several clock mutants was necessary to uncover that different clock genes impact the transcriptome and proteome differently [124]. These observations reinforce the relevance of proteomic studies to have a full understanding of circadian biology.

## 6. Rhythmic Post-Transcriptional Control by Non-Coding RNAs

The role of noncoding genes in several functions related to mRNA metabolism is widely accepted, though their implication in circadian rhythms in plants remains to be elucidated. A tiling array-based study in *Arabidopsis* has shown that around one-fourth of the transcriptome is rhythmic and controlled by the circadian clock. Additionally, several natural antisense transcripts (NATs) showed a circadian behavior [50]. The overall phase distribution of rhythmic NATs tend to be enriched toward the morning, as opposed to sense transcripts, which are preferentially expressed just prior to dawn or dusk [50]. Interestingly, NATs were found for core clock genes such as *LHY*, *CCA1*, *TOC1*, *PRR3*, *PRR5*, *PRR7*, and *PRR9*. This seems not to be a general rule for rhythmic clock genes, as no NATs were found for *GI*, *LUX*, or *ELF3* in this study. Yet, the rhythmic expression patterns of the aforementioned NATs were similar to those of their sense transcript, questioning their relevance in *Arabidopsis* in controlling the transcript abundance of clock genes, at least through typical silencing mechanisms. Recently, *CDF5 LONG-NONCODING RNA* (*FLORE*), a NAT to cycling DOF factor 5 (*CDF5*), has been shown to be a circadianly regulated transcript expressed in an antiparallel manner to *CDF5* [51]. Whereas *CDF5* delays flowering, *FLORE* promotes it by activating the expression of a flowering integrator gene. Still, no changes in clock period were associated with the loss or overexpression of *FLORE*, but it strongly contributed to the circadian oscillation of *CDF5* and other *CDFs* [51]. In *Neurospora crassa*, sense and antisense mRNAs of *frq*, a key component of the circadian clock, were identified in antiphase and were induced by light. Mutant strains in which the induction of the antisense transcript by light is impaired showed a delayed internal clock relative to wild-type strains [125]. So far, this represents the only example providing a link between antisense transcripts and circadian timing.

Other noncoding RNAs that are key post-transcriptional regulators of transcripts are miRNAs. Rhythmic control of miRNA expression appears to be conserved from plants to mammals, though in plants its significance for circadian timing is still unclear. So far, in *Arabidopsis* only *MIR157A*, *MIR158A*, *MIR160A*, *MIR167D*, and *MIR164B* transcripts have been shown to exhibit a circadian expression [50,53,126]. On the other hand, Sire et al. revealed that accumulation of mature miR167, miR168, miR171, and miR398 presented a diurnal oscillation, though they were not controlled by the clock but by environmental cues such as light [53]. Clock performance has not been extensively addressed in mutants of those miRNAs or their targets: Therefore, no genetic evidence has been provided to reinforce the biological significance of these molecular data. Besides, as miRNAs are encoded by gene families, it might well be that clock defects are masked by gene redundancy. Remarkably, the abundance of miR172 appears to be controlled by GIGANTEA (GI) [52], a clock protein linking the circadian pacemaker and photoperiodic flowering in *Arabidopsis* (reviewed in Reference [127]). How GI controls the biogenesis of miR172 remains unknown. MiRNA biogenesis involves their excision from the fold-back precursor. During this process, a double-stranded RNA molecule forming a stem-loop structure is recognized by the miRNA processing machinery to release the mature small RNA [128]. In plants, miRNA biogenesis mutants have not been evaluated for rhythmicity unlike in *Drosophila*, where lack of rhythmicity in such mutants has been observed [129]. Therefore, one cannot rule out that miRNAs could participate in controlling the clock in plants. Nevertheless, no circadian expression has been found for any of the microprocessor genes involved in miRNA biogenesis (Romanowksi and Yanovksy, unpublished). However, rhythmic *At*GRP7 has been shown to bind to some pri-miRNAs *in vivo* and affect their processing into mature miRNAs [130]. Thus, a more systematic study in plants is awaited to resolve these issues.

## 7. Final Remarks

Collectively, the data summarized here demonstrate that circadian expression of several clock mRNAs is fine-tuned by diverse post-transcriptional mechanisms that most likely affect their stability over the course of a day. Thus, the daily rhythmicity in the levels of clock mRNA, a central feature of the circadian clock, may be the result of rhythmic changes in stability. However, transcription seems to be the primary mechanism used to generate rhythmic changes in mRNA levels. Translation of cytosolic mRNAs, post-translational modifications of proteins, as well as epigenetic control, also constitute additional layers of regulation contributing to rhythms and activities of clock proteins or outputs [131].

Our understanding of post-transcriptional networks in plants is still rudimentary in terms of data, as only a few studies have attempted to determine the direct targets of RNA-binding proteins in *Arabidopsis* [132,133,134] and only one has addressed a protein with a regulatory function in circadian timekeeping [87]. However, with genome-wide approaches such as individual-nucleotide resolution crosslinking and immunoprecipitation (iCLIP) now being feasible in plants, the global identification of RNA targets of other circadian RBPs is likely soon to come. This data will not only yield novel insights into post-transcriptional regulation, but will also help to collect information on RNA binding sites for newly developed databases, as previously done for transcription factor binding sites [135]. At the same time, bioinformatics tools need to be continuously updated to assist the analysis of new genomic data, in particular the determination of circadian patterns genome-wide.

Another challenge will be to dissect the molecular function of different splice variants of clock genes. As a first step toward this goal, single mRNA interactome capture can be used, which allows the identification of RBPs associated with a specific transcript. Such methods have already been developed for animal systems [136,137], but not for plants, and have not been applied to clock transcripts yet.

Besides, an important bottleneck in plant circadian biology are genetic screens. Leaf movement rhythms measurements, one of the most robust methods to detect deviations on the circadian period, are at the same time very laborious [138,139,140]. Therefore, the development of techniques that allow high-throughput phenotyping to screen rhythmicity in plants will aid in uncovering new genes involved in the regulation of the circadian clock.

All of these approaches will allow a more detailed inspection of the regulation of the clock, its output genes, and the different layers of regulation, making circadian biology a fascinating and growing field of study.

## Figures and Tables

**Figure 1 genes-09-00616-f001:**
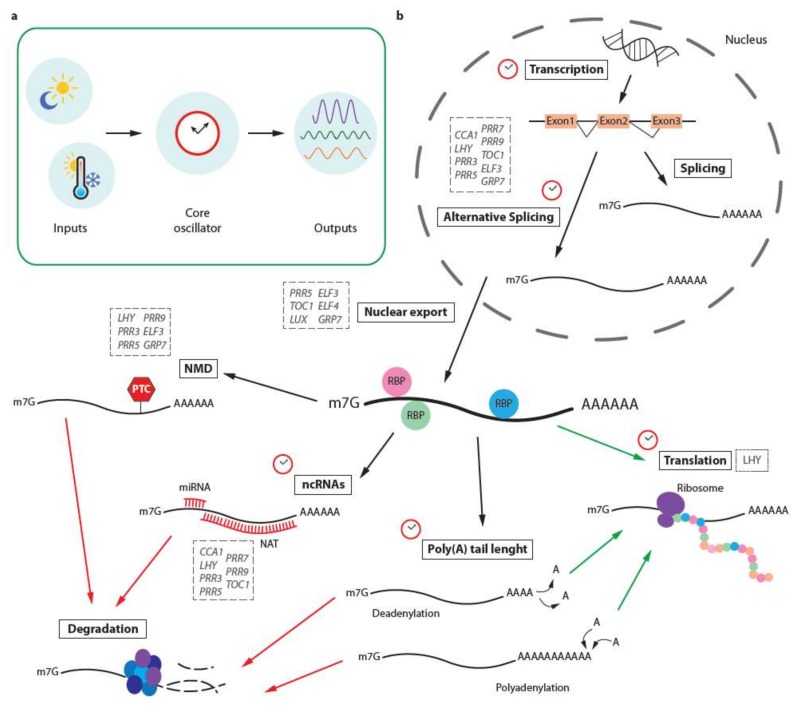
Crosstalk between the circadian clock and post-transcriptional regulatory pathways. (**a**) Environmental signals such as light and temperature (referred to as clock inputs) are integrated by a central oscillator consisting of clock proteins. This core oscillator integrates environmental signals with the endogenous status of the organism and in turn modulates and coordinates a wide range of mechanisms known as “outputs” of the clock. (**b**) Post-transcriptional regulation pathways are some of the outputs known to be circadianly regulated. In this scheme, the clock symbol indicates which of these steps are known to be regulated in this manner. Green and red arrows indicate pathways leading to translation and degradation, respectively. In addition, gray dashed boxes highlight core clock genes known to be subject to post-transcriptional regulation at various steps of RNA processing. ncRNAs: non-coding RNAs; NMD: nonsense-mediated decay; RBP: RNA-binding protein.

**Figure 2 genes-09-00616-f002:**
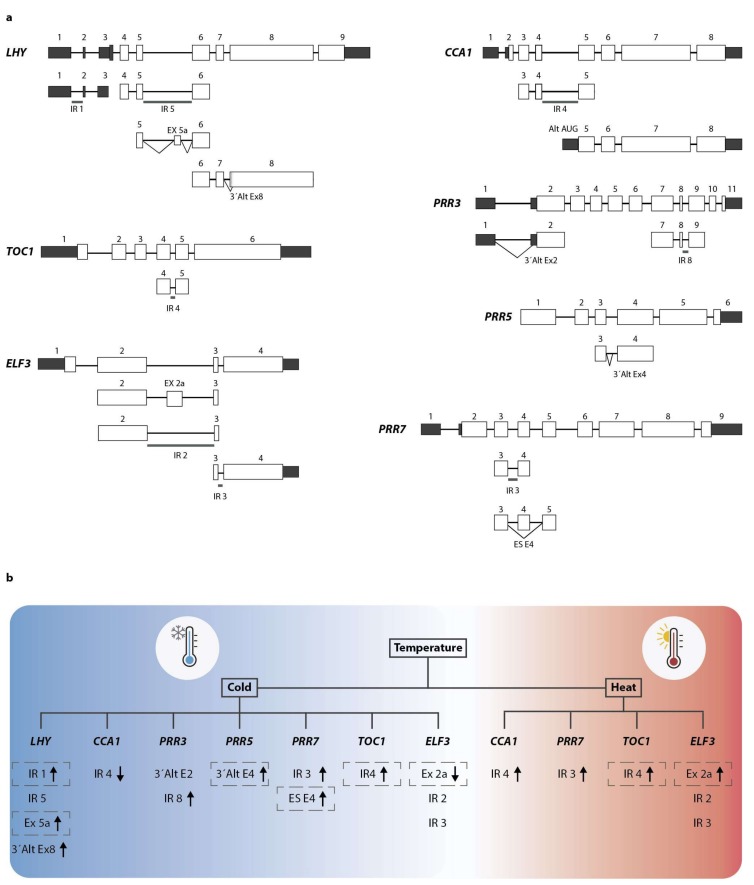
Temperature-regulated alternative splicing (AS) events in core clock genes. (**a**) Genomic structure of major clock components *LHY*, *CCA1*, *PRR3*, *PRR5*, *PRR7*, *TOC1*, and *ELF3*. White boxes depict coding exons and dark-shaded boxes represent 5′ and 3′ UTRs. Reported alternative spliced events are shown below each gene structure. Diagonal lines illustrate splicing events. IR: intron retention; ES: exon skipping; 3′Alt Ex: alternative start of exons. This figure was adapted from References [78,79]. (**b**) Summary of the current knowledge regarding temperature-induced AS of clock genes. Alternative transcripts are divided as cold-regulated or heat-regulated. Arrow lines represent the up- or downregulation of the events under each condition. Gray dashed boxes indicate AS events that result in the introduction of a PTC and consequently NMD.

**Table 1 genes-09-00616-t001:** Genes involved in post-transcriptional mechanisms implicated in clock function.

Mechanism	Gene	Function	Clock-Related Observations	References
Alternative splicing	*PRMT5*	Type II protein arginine methyltransferase of histones, Sm and LSM spliceosomal proteins.	Circadian expression. *prmt5*: Long period; elevated aberrant *PRR9* isoform.	[31,32]
	SM-like genes	Components of the U6 snRNP.	Circadian expression of *LSM5*. *lsm4*and *lsm5*: Long period; changes in expression and alternative splicing of some core clock genes.	[30]
	*SKIP*	Splicing factor and component of the spliceosome.	Role in mediating light input and maintaining temperature compensation of the clock. *skip*: Temperature-sensitive long period; increased aberrantly spliced variants of *PRR7* and *PRR9*.	[38]
	*STIPL1*	RNA-binding protein.	*stipl1*: Long period. Altered accumulation and alternative splicing of core clock transcripts.	[39]
	*GEMIN2*	Spliceosomal assembly factor. Attenuates the effects of low temperature on alternative splicing.	Modulates temperature effects on circadian clock. *gemin2*: Early flowering and short-period phenotype. Alterations in the alternative splicing of core clock genes, similarly affected in wild-type plants at low temperature conditions. Disruption of temperature compensation: Significant period lengthening in response to cold conditions.	[40]
AS-NMD	*AtGRP7* *AtGRP8*	RNA-binding protein. RNA-binding protein.	*AtGRP7* and *AtGRP8* are rhythmic transcripts. *At*GRP7 binding to its own pre-mRNA causes AS followed by NMD, thus autoregulating its expression.	[41,42]
	*TOC1* * *ELF3* *	Core clock gene. Core clock gene.	AS results in PTC and decay. *toc1*: Short period. *elf3*: Arrythmic.	[43,44]
Nuclear transport	*HOS1*	E3 ubiquitin ligase activity. Attenuates cold responses, prevents precocious flowering, and regulates mRNA export.	*hos1*: Early flowering and long-period phenotype. Upregulation of circadian-expressed and core clock genes. Reduction in rhythmicity of circadian clock genes.	[45]
	*AtGRP7*	RNA-binding protein.	*At*GRP7 shuttles between the nucleus and the cytoplasm.	[46,47]
Polyadenylation	Poly(A)-polymerases and deadenylases		Exhibit circadian expression.	[24]
Regulation of translation	*LHY* *	Core clock gene.	Translation is promoted by light, whereas transcription repressed at dusk. Both mechanisms contribute to the narrow peak of *LHY* at dusk. *Lhy*: Short period.	[48]
			Light induces translation.	[49]
nc-RNAs	NATs to clock genes		Natural antisense RNAs to *LHY*, *CCA1*, *TOC1*, *PRR3*, *PRR5*, *PRR7*, and *PRR9 e*xhibit circadian rhythmicity.	[50]
	*FLORE*	CDF5 long-noncoding RNA. Promotes flowering by repressing CDFs.	Antiphasic circadian expression. Reciprocal inhibition contributes to the proper circadian oscillation of both transcripts. Target of PRR7. OxFLORE: Early flowering.	[51]
	miR172	miRNA controlling AP-like transcription factors. Regulates flowering time.	Its biogenesis is controlled by *GI*.	[52]
	*MIR157A*, *MIR158A*, *MIR160A*, *MIR167D*	miRNA controlling diverse developmental processes in *Arabidopsis*.	Transcripts show circadian expression.	[50,53]

* These constitute core-clock genes regulated by post-transcriptional mechanisms. PTC: premature termination codon; miRNA: microRNA; mRNA: messenger RNA, LSM: Like-Sm, snRNP: small nuclear ribonucleoprotein particles, CDFs: CYCLING DOF FACTOR.

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
