# Peer review of "Beyond Transcription: Fine-Tuning of Circadian Timekeeping by Post-Transcriptional Regulation"

_genes, 2018, doi:10.3390/genes9120616_

Round 1

Reviewer 1 Report

The review by Mateos et al. entitled “Beyond transcription – fine tuning of circadian timekeeping by posttranscriptional regulation” revises the current knowledge on posttranscriptional mechanisms regulating, or potentially capable to regulate, the core of the biological clock and the circadian control of molecular and physiological outputs. It is an interesting and clearly written manuscript, focusing on Arabidopsis thaliana. I believe it summarizes the latest research on a very appropriate manner, and highlights the areas of the field that need to be further investigated.

Specific observations and comments:

1.       I believe it would be useful for the reader to specify on the abstract of the manuscript that it is focused on Arabidopsis thaliana.

2.       Introduction, Page 2. The sentence “Through the morning loop, LHY and CCA1 activate the expression of the PSEUDORESPONSE REGULATORS PRR9 and PRR7,…” needs to be corrected: Isabel Carre’s lab recently showed that LHY and CCA1 repress PRR7 and PRR9 expression (Adams S, Manfield I, Stockley P, Carré IA. 2015. Revised Morning Loops of the Arabidopsis Circadian Clock Based on Analyses of Direct Regulatory Interactions. PLoS ONE 10: e0143943).

3.       Line 59. Please review the sentence: “CATALASE3 mRNA oscillations damp to a high level in constant darkness” because it is not clear.

4.       Line 72. U5 is mentioned twice.

5.        On the Section 2, “Posttranscriptional control of the circadian clock by alternative splicing and its connection with environmental responses” might be interesting to mention the paper by Seo et al. (Seo PJ, Park M-J, Lim M-H, Kim S-G, Lee M, Baldwin IT, Park C-M. 2012. A Self-Regulatory Circuit of CIRCADIAN CLOCK-ASSOCIATED1 Underlies the Circadian Clock Regulation of Temperature Responses in Arabidopsis. Plant Cell.). There, the authors show how alternative splicing of CCA1 could modulate its role on the biological clock and cold acclimation. It also provides an example of an alternative transcript coding for a peptide lacking an important domain and therefore, modifying its biological function, instead of the more described here aberrant mRNAs.

In fact, on line 190, the authors state that “AS of CCA1 gives rise to an unproductive isoform retaining intron 4 (CCA1-IR4) that contains a PTC possible leading to NMD (Figure 2A)” whereas it is supposed to originate an alternative translational start site and give rise to an isoform lacking the MYB DNA binding domain (Seo et al, 2012. Plant Cell). I think the authors should revise that sentence.

6.       Line 146. Revise that line for 2 typos/double wording.

7.       Figure 2. The authors describe on the text that ELF3 has 3 isoforms (line 184). However, that is not very clear on Fig. 2.

8.       Paragraph starting on line 275. The authors explain that many rhythmic proteins are derived from non-rhythmic transcripts and conclude that “...indicating that translational control contributes to rhythmic protein abundance [112]”. I believe the possibility of posttranslational regulation contributing to that effect should be added because in fact, it is known that this other layer of control has an important role on protein accumulation, which can shape the observed rhythmic profile.

9.       Line 289. I find the sentence “Rhythmic NATs tend to be expressed in the morning, different to sense transcripts, that are preferentially expressed at dusk [115]” to be confusing, especially considering that the authors mention that NATs were found for LHY and CCA1, two morning genes which peak at dawn. Could they please provide a clarification?

Author Response

The review by Mateos et al. entitled “Beyond transcription – fine tuning of circadian timekeeping by posttranscriptional regulation” revises the current knowledge on posttranscriptional mechanisms regulating, or potentially capable to regulate, the core of the biological clock and the circadian control of molecular and physiological outputs. It is an interesting and clearly written manuscript, focusing on Arabidopsis thaliana. I believe it summarizes the latest research on a very appropriate manner, and highlights the areas of the field that need to be further investigated.

We thank the reviewer for studying the manuscript and giving important feedback. We include below a point-by-point answer to the observations and comments.

Specific observations and comments:

1.       I believe it would be useful for the reader to specify on the abstract of the manuscript that it is focused on Arabidopsis thaliana.

We specified this on the abstract.

2.       Introduction, Page 2. The sentence “Through the morning loop, LHY and CCA1 activate the expression of the PSEUDORESPONSE REGULATORS PRR9 and PRR7,…” needs to be corrected: Isabel Carre’s lab recently showed that LHY and CCA1 repress PRR7 and PRR9 expression (Adams S, Manfield I, Stockley P, Carré IA. 2015. Revised Morning Loops of the Arabidopsis Circadian Clock Based on Analyses of Direct Regulatory Interactions. PLoS ONE 10: e0143943).

We thank the reviewer for this comment. We included this new data of the revised version of the clock and placed LHY as  a repressor of the PRR genes. New citations were included as well.

3.       Line 59. Please review the sentence: “CATALASE3 mRNA oscillations damp to a high level in constant darkness” because it is not clear.

We rephrased. The new version reads:

Along the same lines, CATALASE3 mRNA are kept constantly at high levels in constant darkness, while CATALASE3 promoter-driven luciferase activity still oscillates with an evening peak

4.       Line 72. U5 is mentioned twice.

We changed the text.

5.        On the Section 2, “Posttranscriptional control of the circadian clock by alternative splicing and its connection with environmental responses” might be interesting to mention the paper by Seo et al. (Seo PJ, Park M-J, Lim M-H, Kim S-G, Lee M, Baldwin IT, Park C-M. 2012. A Self-Regulatory Circuit of CIRCADIAN CLOCK-ASSOCIATED1 Underlies the Circadian Clock Regulation of Temperature Responses in Arabidopsis. Plant Cell). There, the authors show how alternative splicing of CCA1 could modulate its role on the biological clock and cold acclimation. It also provides an example of an alternative transcript coding for a peptide lacking an important domain and therefore, modifying its biological function, instead of the more described here aberrant mRNAs.

In fact, on line 190, the authors state that “AS of CCA1 gives rise to an unproductive isoform retaining intron 4 (CCA1-IR4) that contains a PTC possible leading to NMD (Figure 2A)” whereas it is supposed to originate an alternative translational start site and give rise to an isoform lacking the MYB DNA binding domain (Seo et al, 2012. Plant Cell). I think the authors should revise that sentence.

The CCA1 is an interesting case, even mentioned in the commentary from Brown et al, 2015 (doi.org/10.1105/tpc.15.00572) as a special case where AS originate an alternative start site generating a truncated protein with functional importance. The AS even implies IR of intron 4 and consequently a premature stop codon that that could lead to NMD (though not proved). But the presence of an alternative AUG in exon 5 used as start codon generates the truncated protein that has been shown to impair freezing tolerance conferred by the functional CCA1. We added this issue in section 2 citing the Seo et al, paper. In Section 3 where we talk about NMD, we specified that an alternative start site is generated.

6.       Line 146. Revise that line for 2 typos/double wording.

We changed the text.

7.       Figure 2. The authors describe on the text that ELF3 has 3 isoforms (line 184). However, that is not very clear on Fig. 2.

The isoform of ELF3 described in the figure only represents the isoform that changes with environment. Nevertheless, we followed the reviewer comment and changed the text and the figure accordingly, including in the figure all the isoforms of ELF3.

8.       Paragraph starting on line 275. The authors explain that many rhythmic proteins are derived from non-rhythmic transcripts and conclude that “...indicating that translational control contributes to rhythmic protein abundance [112]”. I believe the possibility of posttranslational regulation contributing to that effect should be added because in fact, it is known that this other layer of control has an important role on protein accumulation, which can shape the observed rhythmic profile.

In this review we did not include the topic of posttranslational regulation of splicing, but we rephrased the sentence mentioned above. In the “final remarks” we highlighted the existence of other layers of regulation besides posttranscriptional control, but we do not go into detail in this review. Still we thank the reviewer for the suggestion.

9.       Line 289. I find the sentence “Rhythmic NATs tend to be expressed in the morning, different to sense transcripts, that are preferentially expressed at dusk [115]” to be confusing, especially considering that the authors mention that NATs were found for LHY and CCA1, two morning genes which peak at dawn. Could they please provide a clarification?

We wanted to highlight that the overall phase distribution of NATs tend to be enriched in the morning. NATs tend to be expressed in the morning while sense transcripts are mainly expressed just before dusk or dawn. This picture is seen as a whole, but there are NATs that are expressed outside this morning phase. That is the case for LHY and CCA1 NATs. We made this point clearer by changing the text in the ms.

Reviewer 2 Report

This review by Mateos et al addresses the role of post-transcriptional regulation in the circadian clock mechanism of plants. It is well-written (a few stylistic suggestions are listed below).  It is comprehensive and offers a number of interesting insights and makes a several useful suggestions for further investigation.  In my opinion, this review makes a very useful contribution, going well beyond previous reviews on this topic, providing both deep and informative coverage.  The figures are quite helpful.  I support its publication with only minor modification.

Minor suggestions

Lines 91-93.   I suggest: In Arabidopsis, many PRMT5 methylation substrates are directly involved toin splicing, namelyincluding SmD, SmB, LSM4 and GLYCINE RICH PROTEIN 7 and 8 (At GRP7 and At GRP8) [32].

Line 109 “in the same way as light does” could be misinterpreted as implying mechanistic conservation.  Perhaps modify to read “by changing the phase of circadian oscillations as do abrupt shifts in light, although it is uncertain of the mechanisms by which these shifts are effected are conserved.”

Line 132 “aberrant products can result inhave detrimental effects”

Line 146 “filamentous fungifungus” (singular rather than plural)

Line 152  an allele is not a mutant.  “NMD defective mutants, such as those carrying the strong allele upf1-3,”

Lines 153-157 --  I like this suggestion

Line 257-259  Several decades ago, poly(A) tail length has beenwas connected to circadian regulation in mammals with the observation of daily variation has been foundin the length of the poly(A) tail of vasopressin messenger RNA [102].

Line 267  “likely to be” seems a bit strong—perhaps better to sat “is potentially an important mechanism”

Lines 275-283  The work of Somers showing that ZTL protein cycles although its mRNA does not remains a classic early example of this phenomenon and the authors might wish to consider making it explicit (Kim et al Nature 2007449, 356-360)

I would also suggest adding to this section some discussion of:

Graf, A.; Coman, D.; Uhrig, R.G.; Walsh, S.; Flis, A.; Stitt, M.; Gruissem, W. Parallel analysis of arabidopsiscircadian clock mutants reveals different scales of transcriptome and proteome regulation. Open Biol. 20177, 160333, 10.1098/rsob.160333.

Missra, A.; Ernest, B.; Lohoff, T.; Jia, Q.; Satterlee, J.; Ke, K.; von Arnim, A.G. The circadian clock modulates global daily cycles of mRNA ribosome loading. Plant Cell 201527, 2582-2599, 10.1105/tpc.15.00546.

Line 286 A tiling (not tilling) array ...

Line 301 sense and antisense mRNAs of frq

Line 302 by light is impaired, showed ... – delete the comma

Line 308 „...MIR167D transcripts show a circadian expression“  if MIR trnascripts have not been exhaustively surveryed, it might be better to say „...MIR167D transcripts have been shown to exhibit a circadian expression“  

There is a recent example of a cycling miRNA, miR164, that plays a role in circadian regulation of senescence—Kim et al 2018 PNAs 115(33):8448-8453 

Line 312 „no genetic evidence can reinforce“ „no genetic evidence has yet been provided to reinforce“

Line 350  genetic screenings screens

Author Response

This review by Mateos et al addresses the role of post-transcriptional regulation in the circadian clock mechanism of plants. It is well-written (a few stylistic suggestions are listed below).  It is comprehensive and offers a number of interesting insights and makes a several useful suggestions for further investigation.  In my opinion, this review makes a very useful contribution, going well beyond previous reviews on this topic, providing both deep and informative coverage.  The figures are quite helpful.  I support its publication with only minor modification.

We thank the reviewer for suggestions and useful comments. Answers to each comment is replied below.

Minor suggestions

Lines 91-93.   I suggest: In Arabidopsis, many PRMT5 methylation substrates are directly involved toin splicing, namelyincluding SmD, SmB, LSM4 and GLYCINE RICH PROTEIN 7 and 8 (At GRP7 and At GRP8) [32].

We changed the text.

Line 109 “in the same way as light does” could be misinterpreted as implying mechanistic conservation.  Perhaps modify to read “by changing the phase of circadian oscillations as do abrupt shifts in light, although it is uncertain of the mechanisms by which these shifts are effected are conserved.”

 We changed the text.

Line 132 “aberrant products can result inhave detrimental effects”

We changed the text.

Line 146 “filamentous fungifungus” (singular rather than plural)

We changed the text.

Line 152  an allele is not a mutant.  “NMD defective mutants, such as those carrying the strong allele upf1-3,”

We changed the text.

Lines 153-157 --  I like this suggestion

Thank you.

Line 257-259  Several decades ago, poly(A) tail length has beenwas connected to circadian regulation in mammals with the observation of daily variation has been foundin the length of the poly(A) tail of vasopressin messenger RNA [102].

We changed the text.

Line 267  “likely to be” seems a bit strong—perhaps better to sat “is potentially an important mechanism”

We changed the text.

Lines 275-283  The work of Somers showing that ZTL protein cycles although its mRNA does not remains a classic early example of this phenomenon and the authors might wish to consider making it explicit (Kim et al Nature 2007449, 356-360)

I would also suggest adding to this section some discussion of:

Graf, A.; Coman, D.; Uhrig, R.G.; Walsh, S.; Flis, A.; Stitt, M.; Gruissem, W. Parallel analysis of Arabidopsis circadian clock mutants reveals different scales of transcriptome and proteome regulation. Open Biol. 20177, 160333, 10.1098/rsob.160333.

Missra, A.; Ernest, B.; Lohoff, T.; Jia, Q.; Satterlee, J.; Ke, K.; von Arnim, A.G. The circadian clock modulates global daily cycles of mRNA ribosome loading. Plant Cell 201527, 2582-2599, 10.1105/tpc.15.00546.

We thank the reviewer for this suggestion. We included a short discussion of these issues in the corresponding section.

Line 286 A tiling (not tilling) array ...

We changed the text

Line 301 sense and antisense mRNAs of frq

We changed the text.

Line 302 by light is impaired, showed ... – delete the comma

We changed the text.

Line 308 „...MIR167D transcripts show a circadian expression“  if MIR trnascripts have not been exhaustively surveryed, it might be better to say „...MIR167D transcripts have been shown to exhibit a circadian expression“  

There is a recent example of a cycling miRNA, miR164, that plays a role in circadian regulation of senescence—Kim et al 2018 PNAs 115(33):8448-8453 

We changed the text adding MIR164B as another cycling transcript and add the reference accordingly.

Line 312 „no genetic evidence can reinforce“ „no genetic evidence has yet been provided to reinforce“

We changed the text.

Line 350  genetic screenings screens

We changed the text.